# Role of Base Excision Repair in Innate Immune Cells and Its Relevance for Cancer Therapy

**DOI:** 10.3390/biomedicines10030557

**Published:** 2022-02-26

**Authors:** Shengyuan Zhao, Samy L. Habib, Alireza G. Senejani, Manu Sebastian, Dawit Kidane

**Affiliations:** 1Division of Pharmacology and Toxicology, Dell Pediatric Research Institute, College of Pharmacy, The University of Texas at Austin, 1400 Barbara Jordan Blvd. R1800, Austin, TX 78723, USA; zhaosy@utexas.edu; 2Department of Cell Systems & Anatomy, University of Texas Health, San Antonio, TX 78229, USA; habib@uthscsa.edu; 3Biology and Environment Science Department, College of Art and Science, University of New Haven, 300 Boston Post Rd., West Haven, CT 06516, USA; asenejani@newhaven.edu; 4Department of Epigenetics and Molecular Carcinogenesis, The University of Texas MD Anderson Cancer Center, Science Park, Smithville, TX 78957, USA; mmsebastian@mdanderson.org

**Keywords:** base excision repair, innate immune cells, innate inflammatory signaling, immunotherapy

## Abstract

Innate immunity is critical for immediate recognition and elimination of invading pathogens or defense against cancer cell growth. Dysregulation of innate immune systems is associated with the pathogenesis of different types of inflammatory diseases, including cancer. In addition, the maintenance of innate immune cells’ genomic integrity is crucial for the survival of all organisms. Oxidative stress generated from innate immune cells may cause self-inflicted DNA base lesions as well as DNA damage on others neighboring cells, including cancer cells. Oxidative DNA base damage is predominantly repaired by base excision repair (BER). BER process different types of DNA base lesions that are presented in cancer and innate immune cells to maintain genomic integrity. However, mutations in BER genes lead to impaired DNA repair function and cause insufficient genomic integrity. Moreover, several studies have implicated that accumulation of DNA damage leads to chromosomal instability that likely activates the innate immune signaling. Furthermore, dysregulation of BER factors in cancer cells modulate the infiltration of innate immune cells to the tumor microenvironment. In the current review, the role of BER in cancer and innate immune cells and its impact on innate immune signaling within the tumor microenvironment is summarized. This is a special issue that focuses on DNA damage and cancer therapy to demonstrate how BER inhibitor or aberrant repair modulates innate inflammatory response and impact immunotherapy approaches. Overall, the review provides substantial evidence to understand the impact of BER in innate immune response dynamics within the current immune-based therapeutic strategy.

## 1. Introduction

Innate immunity is triggered in response to pathogen infection or local lesions to promote infection clearance or wound-healing processes [1,2,3]. In addition, innate immune response can be initiated by non-professional immune cells (like epithelial cells, endothelial cells, and fibroblasts) and professional antigen presenting cells (like neutrophils, macrophages, and dendritic cells) [2]. Mechanistically, the activation of innate immune response depends on pattern-recognition receptors (PRRs) that sense danger associated molecular patterns (DAMPs) or pathogen-associated molecular patterns (PAMPs) [4,5]. Pathogen-derived nucleic acids constitute a major class of PAMPs that operate in specific subcellular localizations and recognized by a vast array of PRRs [6]. Due to the abundance of nucleic acids in cells, their corresponding PRRs are evolved to be highly regulated and compartmentalized [7]. Most of the PRRs exhibit high substrate specificity to detect DAMPS [8]. Specific groups of pathogens are recognized via PRRs expressed mainly by cells of the innate immune system. For example, many viruses, bacteria, and intracellular parasites trigger type 1 immunity, with elevations in the expression of specific cytokines [9]. In contrast, multicellular pathogens, including helminths, stimulate a type 2 response, with elevations in IL-4 and IL-13 [10]. Recently published data have also shown that self-released nucleic acids derived from DNA repair deficient cells [1,11] or oxidative mitochondrial DNA damage [2,12] have been considered as DAMPs and trigger innate immune cells response. It is critical to consider both the cell lineage and the specific activation state when assessing the function of a cell of the immune system in response to a specific pathogen or stimuli.

There are different types of innate immune cells such as monocytes, macrophages, neutrophils, and dendritic cells (DC) that can be stimulated by endogenous oxidative stress or exogenous agents such as pathogen [3]. Macrophages originate either from bone marrow-derived monocytes [13,14,15] or from precursor cells derived from the yolk sac or fetal liver during development [16,17,18]. In addition, macrophages play a significant role in maintaining tissue homeostasis, defending against pathogens, and facilitating wound healing [19]. Moreover, macrophages release different types of cytokines, chemokines, and other immune factors to attract other cells to the infection site or tissue injury site [20]. Similarly, monocytes-derived DC also play critical roles in innate immunity [21,22]. DC have PRR and PAMPs, which activate Toll-like receptor (TLR) pathways, type C lectins, release proinflammatory cytokines, and stimulate the innate immune system [23]. Moreover, DCs in the periphery capture and processes antigens, expresses co-stimulatory molecules and produces cytokines, and migrates to lymphoid organs [21]. In addition, DC are equipped with robust oxidative DNA damage repair machinery to maintain its DNA integrity [24]. Neutrophils are another major innate immune cell in the peripheral blood and are produced in the bone marrow from stem cells [25,26]. Neutrophils are recruited to the tumor bed and contribute to further amplifying inflammatory response and releasing metabolites such as hydrogen peroxide that may cause reactive oxygen species (ROS) induced DNA damage [27,28]. Overall, the major common features of neutrophil, monocyte/macrophage activation is the generation of ROS and reactive nitrogen intermediates (RNI), such as O^−2^ and nitric oxide (NO), respectively [29,30,31]. NO is produced by inducible nitric oxide synthase (iNOS) from L-arginine and oxygen, while O^−2^ is catalyzed by NADPH oxidase [32,33,34]. For instance, activated macrophage induced ROS and RNI caused DNA damage, including 8-nitroguanine and 8-oxo-7,8-dihydro-2′-deoxyguanosine (8-oxodG) [35,36]. Those DNA base lesions can lead to base loss or DNA single-strand breaks (SSBs) [37,38,39,40]. Furthermore, cells harbor SSBs progress into S-phase encounter DNA-replication apparatus and convert to double-strand breaks (DSBs) [41,42]. In addition, spontaneous ROS-associated DNA base damage and DSBs can result from RNA transcription, DNA replication, and/or genotoxic stress during infection [43,44,45]. Those DNA damaged sites likely release DNA fragments into cytosolic compartment of the cells and would potentially be recognized as DAMP to trigger innate immune response [43]. Notably, a recent discovery has shown that innate immune cells such as macrophages have two important self-protective mechanisms against oxidative stress associated DNA damage/DAMP-associated innate immune cells activation, including maintaining redox balance via generating antioxidant enzymes such as superoxide dismutase (SOD), glutathione peroxidase (GPx), catalase, and glutathione reductase (GR) [46,47]. The second mechanism is associated with inherent DNA repair pathways to process oxidative DNA damage and restore immune system homeostasis [6,48].

An overall summary of what has been so far discovered and established on DNA damage and repair in innate immune cells biology is provided within this review. In addition, we highlight what remains outstanding and which questions are still unanswered. The key findings of studies that have been carried out in oxidative DNA damage repair linking to innate inflammatory response are provided. The various aspects of BER dynamics to modulate the macrophages population that are present in high numbers in the tumor microenvironment (tumor-associated macrophage) and its role in how BER contributes to cGAS/STING inflammatory pathways is included. Finally, we outline the potential use of some of the DNA repair targeted therapy that modulates innate immune signaling and remodels the inflammatory response.

## 2. Base Excision Repair in Innate Immune Cells

Excessive production of ROS and RNI in innate immune cells, including neutrophils, dendritic cells, eosinophil, and macrophage, leads to oxidative-stress-related DNA base damage, DNA adducts, and SSBs, which may further result in mutations [49,50,51]. BER is the main repair pathway against oxidized DNA bases to maintain genomic integrity [49,52,53,54,55]. Mammalian cells harbor two sub-BER pathways that are dependent on the size of the oxidized DNA base they process and the key enzyme involved in the repair process [56]. The two sub-pathways are known as short-patch BER (SP-BER) and long-patch BER (LP-BER) [57,58]. Short-patch BER engages in repairing one nucleotide gap [44,45], while the long-patch BER involves processing and repairing 2–12 nucleotides gaps. Both BER mechanisms are initiated by DNA glycosylase that recognizes and removes the DNA base lesion. In SP-BER, AP-endonuclease 1 (APE1) cleaved the DNA backbone to generate a 3′-OH terminus at the damage site, followed by DNA polymerase beta adding one nucleotide and the nick sealed by a DNA ligase in a complex with XRCC1 [59]. In comparison, LP-BER utilizes both DNA polymerase beta and other DNA replication enzymes, such as DNA Pol δ and DNA Pol ε, to conduct strand-displacement DNA synthesis. The displaced single-stranded DNA structure or 5′-DNA flap off-load by flap endonuclease I (FEN1) [60] followed by sealing of the DNA nicks by Ligase I or Ligase III [61].

Several studies have shown that neutrophils and monocytes are hypersensitive to accumulated DNA damage derived from oxidative stress due to their insufficient BER function [24,62]. The unrepaired DNA base damage derived from oxidative stress and accumulated BER intermediates in monocytes and triggers DNA damage response (DDR) to halt the cell cycle or induce cell death [50]. Low BER and DSB repair activity observed in neutrophils and monocytes are results from under-expression of BER proteins, including XRCC1, ligase IIIα, ligase I, poly(ADP-ribose) polymerase-1 (PARP1), and repair proteins [25,37,63]. In addition, ROS induces DNA single- and double-strand breaks, leading to activation of the DDR pathways that trigger monocytes apoptosis, whereas macrophages and DCs are resistant to ROS-associated DNA damage and apoptotic cell death [12,24]. In addition, eosinophils harbor efficient DNA repair activity, including BER [62]. Overall, these observations are supported by the presence of a sufficient level of BER proteins, including XRCC1, Ligase I, and DNA polymerase beta (Figure 1) in macrophages, but not in monocytes [50]. In accordance with this, the relevant evidence on how BER factors are also involved in macrophage plasticity is discussed in the next section.

## 3. BER Modulates DNA Damage Induced Innate Immune Inflammatory Response

Several studies have shown that oxidative DNA damage initiates an innate immune inflammatory response via activation of a variety of transcription factors, such as nuclear factor kappa B (NF-κB), STAT, and interferon regulatory factors [51,64,65]. BER function is critical to protect the cells from genomic instability and/or inflammation [37,66]. In particular, BER is expressed in selected types of innate immune cells that may regulate the innate immune response [67]. Few studies highlight the role of oxidative DNA damage repair in innate immune cells; however, detailed mechanistic insight into how loss-of-function BER drives or contributes to DNA-damage-induced inflammatory response remains unknown. The Cancer Genome Atlas (TCGA) data set have shown that somatic BER gene mutations are found in 30% of tumors, and other also indicated that germline variant of BER genes (POLB, APEX1, MUTYH, XRCC1 OGG1, TDG) leads to compromised repair capacity and increases the risk of inflammatory associated diseases, including cancer [68]. Due to impaired function of oxidized DNA damage repair and/or increased accumulation of ROS induced DNA base damage that can trigger DNA damage response (DDR), including activation of Ataxia-telangiectasia mutated (ATM) kinase, ATR, and stimulate nuclear factor kappa B (NF-κB) transcription factors to drive the expression of proinflammatory genes [51,69,70,71]. Moreover, some of the BER proteins such as OGG1 and APE1 can act as transcription factors and are involved in epigenetic regulation of expression of inflammatory cytokines mediated by NF-κB to foster inflammatory response [65,72]. In contrast, BER deficiency (e.g., OGG1) suppresses the expression of proinflammatory genes [72,73,74,75]. However, in some cases, loss of OGG1 function enhances inflammatory response to mitigate bacterial infection [63]. Excessive level of unrepaired DNA damage in the nucleus- and factors-associated DNA end resection during double-strand break repair, including the Bloom syndrome (BLM) helicase and exonuclease 1 (EXO1), plays a major role in generating DNA fragments, and the cytoplasmic 3′–5′ exonuclease Trex1 is required for their degradation. However, the lack of DNA degrading nucleases enzyme in the cytosol likely contributes to increasing cytosolic DNA [43,76]. The cytosolic damaged DNA is recognized by the innate immune system mediated by different types of receptors or DNA sensors [77,78], which subsequently triggers inflammatory immune response [79,80,81]. Moreover, DNA fragments released from apoptotic cells [82,83] can activate innate immune cells in a Toll-like receptor 9 (TLR9) [84] and cGAS/STING dependent fashion [2].

Multiple studies have shown that BER prevents DNA damage derived from oxidative stress in innate immune cells and modulates the inflammatory response [73]. BER deficiency leads to accumulated intrinsic oxidized DNA bases, BER intermediates (AP sites, SSBs), and DNA-replication-associated DSBs [85]. Alternatively, DNA damage derived from oxidative stress leads to activation of DNA damage response (DDR) and modulate inflammatory response [67,86]. DDR signaling enhances the DNA repair capacity of the cells by inducing DNA repair transcriptionally or post-translation modification to counter DNA damage [87]. However, several BER knockout mouse models are found to develop inflammatory associated diseases, including cancer and autoimmunity [1,88,89]. For instance, mice are susceptible to inflammation when they harbor aberrant DNA glycosylases function (e.g., OGG1, MUTYH, NEIL2, and NEIL3) since they are unable to remove oxidized DNA bases [66,90,91,92,93,94]. Moreover, mouse carrying mutation in dRP lyase domain of DNA polymerase beta displays higher innate immune inflammatory response [37]. Another study found that a nuclease-deficient Flap endonuclease-1 (FEN1) mutant mouse model accumulates apoptotic-associated cytosolic DNA that forms complex with IgG to promote chronic inflammation [95,96]. However, it is unknown whether the cytosolic DNA in BER-deficient cells is predominantly generated from DNA damage product from nuclei or mitochondrial DNA that likely activates innate immune signaling and promotes the inflammatory response. Recent work has implicated chromosol instability contribute to inducing innate immune inflammatory pathways [97]. Detection of DNA in the cytoplasmic compartment of the cells induces a type I interferons (IFNs) response to amplify the innate immune signaling-based inflammatory response [89]. DNA leakage from damaged nuclear and/or mitochondrial DNA has the potential to induce innate immune signaling through binding to a cyclic guanosine monophosphate (GMP)-adenosine monophosphate (AMP) synthase (cGAS) DNA sensor [98]. These products catalyze the production of cyclic GMP-AMP (cGAMP) and further lead to the activation of stimulators of interferon genes (STING) [98]. Previously, other studies have shown that cGAS and STING have been identified as intracellular DNA sensors that activate the interferon pathway in response to virus infection and intercellular pathogens [81,99]. STING is a signaling molecule, often located at the endoplasmic reticulum (ER), and it is essential for controlling the induction of type 1 interferons (IFN) and inflammatory mediators [100,101,102]. In addition, interferon-inducible protein 16 (IFN16) binds to dsDNA and promotes STING dependent activity [100,103,104] of TBK1 phosphorylation that subsequently leads to activation of transcription factors interferon regulatory factor 3 (IRF3) and Nuclear factor-kappa B (NF-kB), which triggers innate immune gene transcription [76,88,89,105,106]. In particular, macrophages harbor robust DNA-sensing activity [107]. Macrophage has the ability to activate intrinsic cGAS/STING pathways due to persistence of endogenous oxidative stress-associated DNA base damage and promote proinflammatory response [108]. Alternatively, dsDNA released from oxidative stress-associated DNA damage product may also trigger activation of cGAS/STING a (Figure 2). Moreover, exposure to exogenous DNA damaging agents such as radiation and chemotherapy (e.g., paclitaxel, BSO) triggers the death of tumor cells and the production of oxidized DNA base damage, which activates innate immune cells, including macrophage and dendritic cells via the STING-dependent signaling pathway [43,109].

## 4. Role of BER in Macrophage Plasticity

As an essential component of innate immunity, macrophages can inhibit or promote cell proliferation and tissue repair [110]. Macrophages originate from peripheral blood monocytes and harbor heterogeneous phenotypes [17]. Moreover, macrophages can adjust their phenotype in response to various extracellular signals [111,112]. M1 macrophages secrete high levels of proinflammatory cytokines and increased concentrations of superoxide anions (O_2_^−^), oxygen radicals, and nitrogen radicals to increase their cytotoxicity activity, as required in early phases of tissue repair [113]. Conversely, M2 macrophages produce an extracellular matrix and reduce proinflammatory cytokine levels to promote tissue repair [100]. More specifically, tumor-associated macrophages (TAMs) account for more than 50% of tumor-infiltrating cells in the tumor microenvironment (TME) [114,115]. TAM is derived from two different sources including tissue-resident macrophage and macrophage derived from monocytes [101,102]. TAM is the most abundant cell type among the tumor-promoting stromal cells in solid tumors [103,104]. TME consists of cytokines/chemokines produced by tumor cells or stroma cells, and it is critical to shaping the macrophage functional plasticity [116,117,118]. In particular, macrophage chemoattractant molecules [e.g., C-C Motif Chemokine Ligand 2 (CCL2)/monocyte chemoattractant protein 1 (MCP-1), colony-stimulating factor 1 (CSF-1)] play a major role to recruit TAMs into the tumor microenvironment. TAMs are categorized into M1 and M2 phenotypes [119]. The switch between M1 (anti-tumorigenesis) and M2 (pro-tumorigenesis) is known as “macrophage polarization” [120]. The relationship between TAMs and cancer cells has been studied intensively [121,122], and it is well established that TAMs play a major role in tumor development and progression [123,124]. Several studies have shown that increased TAM infiltration negatively correlated with the overall survival of cancer patients [125,126,127]. TAMs promote tumor progression via enhancing cancer cell genetic instability, facilitating cancer stem cells replication and survival, supporting metastasis, influencing adaptive immunity response, and promoting chemoresistance [128]. Specifically, M2 macrophages play key roles in cancer development by promoting angiogenesis in tumors and accelerating metastasis, as well as fostering resistance to different cancer therapy [128,129,130]. In addition, M2 macrophages exhibit an anti-inflammatory response to promote tumor progression, while M1 macrophages support proinflammation response hostile to tumor cells. In contrast, finding from several preclinical studies demonstrated that infiltration of M1 in the tumor microenvironment is associated with better overall survival of patients [126].

Genetic variants of BER genes may result in differences in immune system components, including the abundance and activation states of immune-cell types, the expression of immunomodulatory molecules, and regulation of immune-related genes [131]. Several studies have identified somatic and germline variants of BER genes and have provided mechanistic insight their impact in tumor initiation [132,133,134,135,136]. Germline genetic variation in genes that code for DNA repair may influence an individual’s innate immune function. Moreover, recent studies have shown that germline may influence which mutations are selected for in the growing tumor [137,138]. In contrast, few studies have shown that macrophage carrying DNA repair defect accumulates DNA damage that derives metabolic reprogramming and eventually leads to chronic inflammation [139]. However, it is unknown that human germline variant of BER genes impacts macrophage functional plasticity. In comparison, altered BER function in cancer cells may contribute to unbalanced recruitment of macrophages in the tumor microenvironment. There are several mouse models established with altered BER functions to understand mechanisms of inflammation-associated cancer pathogenesis. For example, MUTYH knockout mice demonstrated that BER has a capacity to modulate inflammatory response via macrophages infiltration in the tumor microenvironment [103]. Furthermore, mouse models studies provide mechanistic insight into how TAMs are recruited to primary tumors and metastatic tumors [140]. In line with this, a high number of infiltrations of M2 macrophage in the TME is associated with poor overall survival of patient and worse disease outcome [141,142]. In addition, BER genes, including MPG and NEIL2, are overexpressed and have a positive correlation with the infiltration of M2 macrophage in the tumor microenvironment (TIMER 2.0, CIBERSORT) (Figure 3). However, experimental evidence has shown that increased TAM infiltration enhances the DNA repair capacity of the tumor cells and fosters low treatment response [143]. Furthermore, treating tumor with oxidative DNA damaging chemotherapeutic drug, such as Paclitaxel, promotes the rewire of TAMs toward tumoricidal phenotypes through activation of TLR4 [144].

Different strategies to target TAMs have been proposed including depleting and/or reprogramming TAMs to improve clinically approved chemotherapeutic drugs response [145,146]. Therefore, using TAM as a therapeutic target in the tumor microenvironment will likely provide a novel approach to prevent the progression of solid tumors and prevent metastatic disease [128]. One of the therapeutic strategies focused on targeting BER genes such as APE1 and PARP1 promotes attracting antitumor macrophage in tumors microenvironment and enhancing antitumor immune response [103,119,147,148,149]. In particular, targeting the Colony Stimulating Factor1 (CSF1)–Colony Stimulating Factor1 Receptor (CSF-1R) axis has received significant consideration for DNA repair targeted therapeutic purpose to overcome immunosuppressive macrophages [150]. CSF1 (also known as M-CSF) plays a critical role in promoting the differentiation of monocyte to macrophage lineages [151]. CSF1 is abundantly expressed by several tumor types; this ligand-receptor pathway has been extensively investigated in tumor models and constitutes a paradigm of TAM-cancer cell interaction [1,152]. Moreover, DNA damaging agents such as radiation therapy induce the expression of CSF1, which results in CSFR1 dependent infiltration of immunosuppressive macrophages to the TME [153,154]. CSF-1R is expressed by macrophages and provides a potential target to eliminate TAM. Overexpression of CSF-1, the major lineage regulator for macrophages, is associated with poor prognosis in breast, ovarian, endometrial, prostate, hepatocellular, and colorectal cancer [15], as the intra-tumoral presence of CSF1R^+^ macrophages correlates with poor survival in various tumor types [127,155]. Moreover, our in silico analysis shows that overexpression of CSF1 and CSFR1 is positively correlated in tumors that harbor high expression of DNA repair genes such as APEX1 and PARP-1 (TCGA, TIMER 2.0, CIBERSORT). In line with this, exploiting the CSF1–CSF1-R axis of macrophage and DNA repair targeted therapy may likely provide an alternative platform to enhance therapeutic efficacy and may improve overall patient survival.

## 5. Targeting PARP1 Modulate Macrophage Mediated Tumor Inflammatory Response

Poly(ADP-ribose) polymerase-1 (PARP1) contributes to many physiological processes, including DNA replication and transcription, DNA repair, regulation of cell differentiation, and apoptosis [156]. PARP1 plays a significant role in BER pathway to repair SSBs [157]. In addition, PARP1 is also critical to repair replication-associated DNA damage during DNA synthesis [157,158,159]. Apart from its primary role in genome integrity maintenance, PARP1 appears to protect M1 macrophages from oxidative-stress-related cell death through transcriptionally regulating antioxidants such as MnSOD (SOD2), glutathione reductase, and thioredoxin reductase [160,161]. In addition, the level of PARP1 is reduced when macrophage is stimulated with lipopolysaccharide (LPS) and polarized to pro-M1 phenotype [160]. Further, PARP1 modulates macrophage polarization via inflammatory mediators such as high-mobility group box protein 1 (HMGB1) [162]. HMGB1 is a non-histone nuclear proteins and very potent inflammatory mediator [144,163]. HMGB1 is secreted by activated innate immune cells, including monocytes and macrophages [164], and is also released by necrotic cells [165,166]. In the setting of immune challenge, PARP1 PARylates HMGB1 promotes its acetylation, detaches from chromatin, and thus allows the translocation of HMGB1 from the nucleus to the cytoplasm [164,167]. Subsequently, the cytoplasmic HMGB1 can be secreted into the extracellular space as DAMP to induce inflammatory mediator such as cytokine and chemokine activities [168]. Moreover, HMGB1 interacts with the receptor for advanced glycation products (RAGE) and leads to macrophage polarization into M1 phenotype [169], whereas HMGB1 interacts with C1q complement to induce M2 polarization [170]. Conversely, the TCGA dataset shows that expression of PARP1 in tumors negatively correlated with type I interferon gene expression (CCL5, IFNβ, ISG15). The recent results reviewed in this article indicate that PARP1 inhibitors, such as Olaparib treatment in cancer cells, reduce polarization of adjacent macrophage to M2 phenotype via accumulation of unrepaired DNA damage and BER intermediates [150]. Moreover, PARP1 inhibition is associated with increased levels of cytosolic DNA, which can trigger cGAS-STING pathways to activate IRF3 and NF-κB dependent expression of several genes that mediate innate immune response [171,172]. This includes expression of type 1 interferons and T-cell-recruiting chemokines [C-C Motif Chemokine Ligand 5 (CCL5), C-X-C Motif Chemokine Ligand 10 (CXCL10)], leading to a higher level of tumor-infiltrating T-cells [171]. Additionally, PARP1′s role in sensing DNA damage mediates a non-canonical pathway of STING activation [173]. Upon binding DSBs, PARP1 recruits and activates ATM, which subsequently activates the ubiquitin ligase TRAF6 to translocate to the cytosol and interact with IRF16, which results in STING activation [174]. This alternative pathway stimulates STING-dependent activation of the transcription factor NF-κB to generate inflammatory response [175]. It is possible that PARP1 inhibitor exacerbates accumulation of unrepaired DNA damage in cells that harbor pathogenic germline variant of BER that caused impaired BER function (Figure 4). Overall, targeting PARP1 will likely stimulate cGAS/STING dependent on the innate immune signaling and enhance antitumor immune response that potentially contributes to maximizing immunotherapy treatment efficacy (Figure 4 labeled with blue).

## 6. Targeting APE1 Modulate Macrophage Mediated Tumor Inflammatory Response

Apyrimidinic endonuclease 1/redox factor-1 (APE1/Ref-1) is a multifunctional enzyme participating in both oxidative DNA damage repair and redox signaling in cancer [176,177]. APE1 enhances the affinity of transcription factors (e.g., activator protein 1 [AP-1], nuclear factor-κappa B [NF-κB], p53, and others) binding to DNA to modulate inflammatory response [178]. APE1 is overexpressed in several types of cancer and alters treatment outcomes [149,178,179,180,181]. Previous studies have shown that APE1 regulates innate inflammatory response in macrophages [182]. APE1 overexpression in tumors is associated with poor prognosis in cancer patients and upregulation of immunosuppressive expression of genes [e.g., program death ligand-1 (PD-L1)] [183]. In addition, APE1 overexpressed tumor positively correlates with pro-tumor M2 macrophage infiltration and is associated with poor overall survival (CIBERSORT, TIMER 2.0). In line with this, the release of inflammatory cytokines/chemokine by M2 macrophage played a major role in cancer promotion and progression [121], whereas anti-inflammatory drugs reduce the risk of cancer [184]. Several studies have suggested the potential of APE1 as a therapeutic target in cancer [178]. However, little has been investigated on the role of APE1 in regulating innate immune cells mediated antitumor inflammatory response. Furthermore, APE1 expression in tumors negatively correlated with Type I interferon genes expression such as IFNB, CCL5 expression (TCGA) implicated that its overexpression contributes to suppressing the antitumor immune response. However, a limited number of studies have shown that targeting APE1 with inhibitor (E3330) reduces macrophage-mediated NF-κB and AP-1 signaling, which leads to low expression of inflammatory mediators such as TNF-α, IL-6, IL-12, NO, and PGE_2_ [185,186,187]. That experimental evidence suggests that targeting APE1 (E3330) in cancer and innate immune cells likely provides an alternative therapeutic strategy to stimulate antitumor immunity and may also increase anti-PD-L1 therapeutic response in cancer (Figure 4, labeled with purple).

## 7. Exploiting BER as Innate Immune Modulator for Cancer Therapy

BER modulates the immune response in inflammation-associated human disease [188,189]. Inhibition of the BER pathway promotes the accumulation of oxidative-stress-related DNA damage in an acidic TME [190]. There are different types of immune-based therapy for cancer; however, the therapeutic efficacy depends on the interaction of cancer cells with different components of the TME, including the DNA repair capacity, the density of infiltration of tumor-associated macrophage, and the status of innate immune signaling. In the next section of this review, we discuss the potential contribution of some of the tumor-associated functions of DNA repair proteins and their role in the context of immune-based therapy. We discuss the two important scenarios that are associated with defective DNA damage repair and activation of immune response to enhance immune-based therapy.

(i) Exploiting BER defect associated with reactivation of cGAS/STING in TME. Several types of cancer cells are found to silence their intrinsic cGAS-STING signaling pathways in the TME by epigenetic hypermethylation [191]. Loss of cGAS-STING signaling facilitates tumor cells to escape immune surveillance, thereby promoting carcinogenesis and resistance towards immunotherapy. Further studies in mouse models have shown that loss of cGAS-STING pathway causes low expression of type I IFN genes, which leads to a low number of infiltrating CD3^+^ CD8^+^ T cells [192]. Recent evidence emerged on activating “cold tumor” to “hot tumor” through the restoration of cGAS/STING pathway to enhance immune-based therapy response. One viable strategy is to target the BER pathway. For example, PARP1 inhibitor impairs DNA repair capacity and enhances the accumulation of PARP1-DNA complex trapping and further triggers the release of cytosolic DNA for activation of cGAS/STING [162,193]. There have been multiple attempts in developing a number of synthetic STING agonists, including Cyclic dinucleotides (CDNs) (e.g., MK-1454) and non-CDN small molecule compounds (MK2118, SR717, TK676, MSA-1, and MSA-2) to restore activation of cGAS/STING pathways [194]. Further, divalent metal ion such as manganese ion plays an important role during oxidative stress and act asco-factor for many DNA repair enzymes, including in BER factors for activation of cGAS/STING signaling pathways. A newly discovered mechanism suggests manganese released from mitochondria to the cytosol activates cGAS/STING pathways [195] and enhances immunotherapy response [196].

(ii) Inhibition of BER factors enhances program cell death ligand-1 (PD-L1) expression in cancer cells. Cancer-specific BER defects are abundant in malignant tissues [197,198,199,200]. Immune checkpoint therapy has recently emerged as a promising next-generation cancer treatment. Several studies have shown that the PD-L1 level of expression in tumors is an important factor to influence the therapeutic efficacy of response of cancer patients [201,202]. Interestingly, traditional chemotherapeutic agents inducing DNA damage have been found to upregulate the expression of PD-L1 in many cancer types [203]. Furthermore, emerging evidence suggests that defects in DNA repair machinery lead to upregulation of PD-L1 [204]. Recently, Permata et al. showed that BER gene expressions are negatively correlated with PD-L1 expression in tumors and oxidative DNA damaging agents exacerbate the expression of PD-L1 [205]. Loss of BER that enhances the accumulation of SSBs and progress into the S phase of the cell cycle likely causes replication-associated DSBs in cancer cells [85,203]. BER defects/low expression show high microsatellite instability increased neoantigen production and PD-L1 expression in tumors [206]. In the future, exploring BER deficiency and oxidative stress DNA damage associated upregulation of PD-L1expression in tumors will likely provide an additional immune biomarker to introduce the immune-based therapeutic strategy.

## 8. Conclusions and Future Direction

In this review article, we presented a new direction to highlight the cross talk of DNA repair capacity of cancer cells and innate immune cells phenotypes. The overall mechanistic insight into the DNA repair capacity of cancer cells and innate immune cells can provide a fundamental framework to develop a therapeutic strategy that considers the innate immune cells’ response in the tumor microenvironment. Even though several BER germline and tumor-associated mutations were uncovered from different studies [68,135,198,200], their functional impact on innate immune cells such as macrophage and dendritic cells is unknown. In particular, macrophages phenotypes and functional plasticity are influenced by several factors, including oxidative DNA damage and repair capacity. Based on our in silico analysis, the positive correlation of overexpression of BER genes in tumor cell versus infiltration of M2 macrophage impacts the overall survival of patients. Within this context, and similar to other studies, we have shown that reprogramming M2 macrophage into an M1-like phenotype is a potential cancer therapeutic strategy [207,208]. Further, our comprehensive review can inform future studies to consider BER pathways as a potential target to modulate the innate immune signaling and promote the inflammatory response. The future may uncover potential mechanistic insight into how the DNA repair pathway modulates the innate immune response in terms of reprogramming the tumor microenvironment, restoring antitumor immunities, and enhancing cancer immunotherapy treatment response.

## Figures and Tables

**Figure 1 biomedicines-10-00557-f001:**
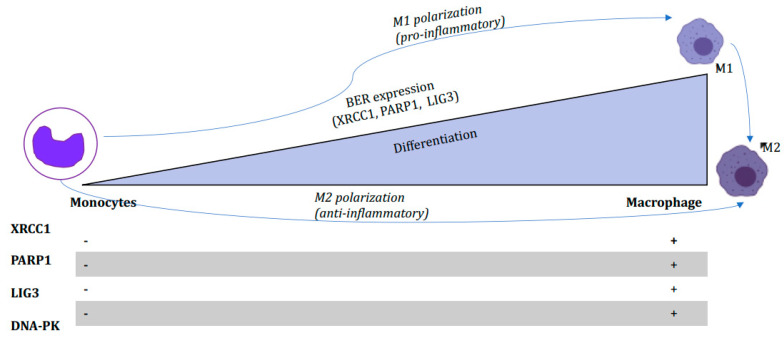
Base excision repair capacity in macrophage. Monocytes are key players in the immune system and contribute as a source for dendritic cells and macrophages. Monocytes differentiate into M1 and M2 macrophages. DNA repair capacity restored in M1 and M2 macrophage. Monocytes are deficient in main base excision repair genes, including POLB, XRCC1, and ligase III and non-homologous end joining repair genes, including DNA-PK.

**Figure 2 biomedicines-10-00557-f002:**
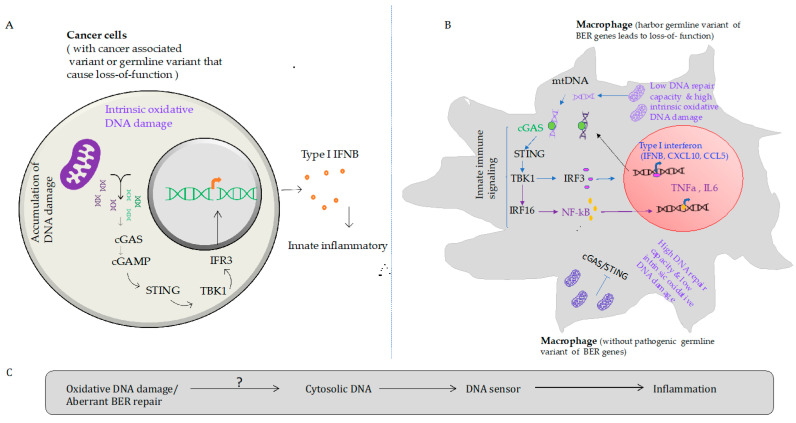
Aberrant BER induces DNA-sensor-mediated inflammatory signaling. The cGAS-STING signaling pathway in BER deficient tumor cells (**A**) and immune cells (**B**). In both cases, tumor-associated mutation of BER genes or germline variant associated with impaired BER function in cancer cells and/or macrophage may contribute to accumulation of unrepaired endogenous oxidative DNA damage product that may leak to cytosol and stimulate cGAS and produce second messenger 2′,3′-cGAMP. After 2′,3′-cGAMP binds to STING protein and then phosphorylate TBK1 that leads to IRF3 activation and/or NF-κB to facilitate their translocation into the nucleus and drives the expression of inflammatory genes, including Type I interferon. In contrast, macrophages with high DNA repair capacity (without pathogenic variant of BER gens) suppress innate immune signaling mediated by cGAS/STING. (**C**) Schematic representation of impaired BER function will likely leads to accumulation of oxidative stress related DNA damage and may trigger the release of cytosolic DNA and cGAS/STING dependent inflammatory response.

**Figure 3 biomedicines-10-00557-f003:**
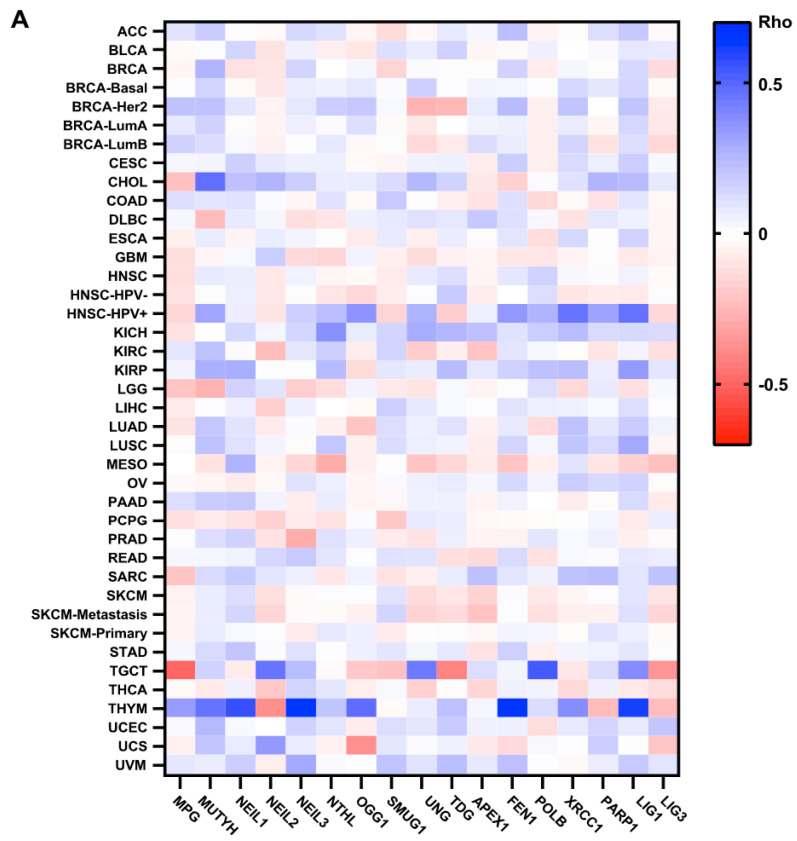
BER gene landscape and its impacts on macrophage infiltration in tumor microenvironment. (**A**) Heatmap of overexpression of BER genes from 40 types of cancer and 17 different BER repair genes expression in tumor cells versus macrophage infiltration. (**B**) Five representative genes (NEIL2, OGG1, APEX1, POLB, DNA LIG3) selected to demonstrate the correlation of overexpression of those repair genes and M2 infiltration in tumor microenvironment. Rho represents correlation coefficient. All data were extracted from TCGA data set and analyzed using TIMER 2.0 using CIBERSORT.

**Figure 4 biomedicines-10-00557-f004:**
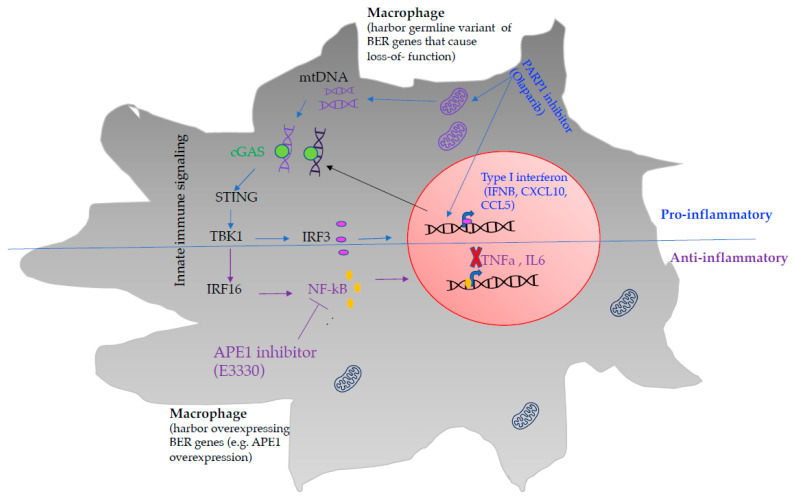
BER-based therapeutic intervention in immune cells (macrophage) modulates inflammatory signaling. Macrophages harbor pathogenic germline variant of BER genes treated with PARP1 inhibitor (Olaparib) will likely accumulate unrepaired products such as PARP1-DNA complex and generate cytosolic DNA to stimulate cGAS/STING pathways and enhance type interferon response (represent by blue color). In contrast, macrophages harbor overexpression of BER genes (e.g., APE1) targeted by APE1 inhibitor (E3330) suppresses NF-kB mediated inflammatory response (represented by purple color).

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
