# Peer review of "Role of Base Excision Repair in Innate Immune Cells and Its Relevance for Cancer Therapy"

_biomedicines, 2022, doi:10.3390/biomedicines10030557_

Round 1

Reviewer 1 Report

The manuscript "Role of Base Excision Repair in Innate Immune Cells and Its Relevance for Cancer Therapy", written by Zhao S, Habib SL, Senejani A, Sebastian M and Kidane D, is a review describing the role of base excision repair in macrophages and related cells.

The manuscript is bed written (I detected more than 80 grammatical or syntax mistakes).

Already in the Abstract it is not clear whether the topic is oxidative stress and its repair enzymes present in the macrophages or it is the oxidative stress which immune cells can produce in cancer cells.

Many statements are not correct or need references:

"germline and somatic mutations are abundant in human cells causes loss of DNA repair function"...

"accumulation of DNA damage leads to chromosomal instability and activate the innate immune signaling"

The manuscript consists of the Introduction and the description of the BER in innate immune cells, BER defects which contribute to innate immune response, roles of BER in macrophage plasticity and presentation of the possibilities for BER exploitation in cancer therapy.

In the Introduction, basic principles of innate immunity and cell recognition of pathogen antigens are

presented. Several statements need more detailed explanation:

"self-released nucleic acids derived from DNA replication stress" – this applies only on cells deficient in certain molecules.

It is not explained that cells, producing oxidative species to protect the organism, do this in their environment or in phagosomes, and not in their own cytoplasm. And then their "aim" is to kill the cell. When describing macrophages inducing DNA damage by ROS they do this in neighboring cells – and this is not clear from the text.

At the end of the Introduction, tumor-associated macrophages are mentioned, without explanation of their biology.

In the second paragraph, base excision process is described. Kaina et al. showed that monocytes do not have high expression of BER enzymes, but concluded that they undergo apoptosis when damaged. It is not clearly described.

In the third paragraph, the topic is BER proteins deficiency in tumors. The authors say that excessive level of unrepaired DNA base damage in the nucleus and lack of DNA degrading nucleases in the cytosol likely contribute to increase in cytosolic DNA. The reference is an article on autoimmune disease cells. More detailed explanations are needed for the statements.

Further on, cGAS-STING mechanism is mentioned, but without detailed explanation, it could be understood that it normally reacts with damaged nuclear DNA leaked from the nucleus. It is not explained that this system recognizes pathogen DNA.

Figure 3 presents macrophage, as a BER deficient immune cell: Previously it was presented that macrophages are not BER deficient.

Also, the data in the second part of the manuscript are not clearly described, mechanisms are superficially mentioned, what can lead to misunderstanding. The role of PARP in BER is not mentioned, role of HMGB1 needs more detailed explanation, more details on tumor-associated macrophages are needed, etc.

Author Response

The manuscript "Role of Base Excision Repair in Innate Immune Cells and Its Relevance for Cancer Therapy", written by Zhao S, Habib SL, Senejani A, Sebastian M and Kidane D, is a review describing the role of base excision repair in macrophages and related cells.

The manuscript is bed written (I detected more than 80 grammatical or syntax mistakes).

Response: We have now  revised the manuscript  and  is edited with professional.

Already in the Abstract it is not clear whether the topic is oxidative stress and its repair enzymes present in the macrophages or it is the oxidative stress which immune cells can produce in cancer cells.

Response:We have  clearly stated the macrophage generated ROS effect  on other neighboring cells  and also possibility of self- induced oxidative stress  in macrophage carrying DNA repair defects.  

Many statements are not correct or need references:

Response: We have included the references.

"germline and somatic mutations are abundant in human cells causes loss of DNA repair function"...

Response:  We have now included relevant references that represent  germline variant that cause loss-of function and increase predispose for cancer and other  inflammatory disease in the revised manuscript section.

"accumulation of DNA damage leads to chromosomal instability and activate the innate immune signaling"

Response: We have now included references

The manuscript consists of the Introduction and the description of the BER in innate immune cells, BER defects which contribute to innate immune response, roles of BER in macrophage plasticity and presentation of the possibilities for BER exploitation in cancer therapy.

In the Introduction, basic principles of innate immunity and cell recognition of pathogen antigens are presented. Several statements need more detailed explanation:

Response: We have now included the detail of specific groups of pathogens that are recognized via PRRs expressed mainly by cells of the innate immune system. For example, many viruses, bacteria and intracellular parasites, trigger type 1 immunity, with elevations in the expression of specific cytokines, including interleukin ( IL-17) and interferon-γ (IFN-γ). In contrast, multicellular pathogens, including helminths, stimulate a type 2 response, with elevations in IL-4 and IL-13.  It is thus important to consider both the cell lineage and the specific activation state when assessing the function of a cell of the immune system in response to a specific pathogen or stimuli.

"self-released nucleic acids derived from DNA replication stress" – this applies only on cells deficient in certain molecules.

Response: We have now included the specific DNA repair defect and provide relevant references.

It is not explained that cells, producing oxidative species to protect the organism, do this in their environment or in phagosomes, and not in their own cytoplasm. And then their "aim" is to kill the cell. When describing macrophages inducing DNA damage by ROS they do this in neighboring cells – and this is not clear from the text.

Response: We have now made it clear that macrophage released ROS has a potential to induce damage on the neighboring cells  including cancer cells  and monocytes.  

At the end of the Introduction, tumor-associated macrophages are mentioned, without explanation of their biology.

Response: We have now included tumor associated macrophage represent high number in  the microenvironment of solid tumors.

In the second paragraph, base excision process is described. Kaina et al. showed that monocytes do not have high expression of BER enzymes, but concluded that they undergo apoptosis when damaged. It is not clearly described.

Response: We have now included that  ROS induces DNA single- and double-strand breaks leads to activation of the DDR pathways that triggers apoptosis in monocytes, whereas macrophages and DCs are protected from  apoptotic cell death.

In the third paragraph, the topic is BER proteins deficiency in tumors. The authors say that excessive level of unrepaired DNA base damage in the nucleus and lack of DNA degrading nucleases in the cytosol likely contribute to increase in cytosolic DNA. The reference is an article on autoimmune disease cells. More detailed explanations are needed for the statements.

Response: We have now provided additional cancer relevant references and included detail of the DNA end resection enzymes including the Bloom syndrome (BLM) helicase and exonuclease 1 (EXO1), play a major role in generating DNA fragments and that the cytoplasmic 3′–5′ exonuclease Trex1 is required for their degradation. We have also described the potential impact of loss of cytosolic exonuclease (Trex1).

Further on, cGAS-STING mechanism is mentioned, but without detailed explanation, it could be understood that it normally reacts with damaged nuclear DNA leaked from the nucleus. It is not explained that this system recognizes pathogen DNA.

Response: We have included information the role of cGAS/STING in protection of host against pathogens including viral infection and intercellular pathogens.

Figure 3 presents macrophage, as a BER deficient immune cell: Previously it was presented that macrophages are not BER deficient.

Response: We have now modified Figure 2 and represented BER deficient cancer cells and macrophage that harbor germline variant of BER genes.

Also, the data in the second part of the manuscript are not clearly described, mechanisms are superficially mentioned, what can lead to misunderstanding. The role of PARP in BER is not mentioned, role of HMGB1 needs more detailed explanation, more details on tumor-associated macrophages are needed, etc.

Response: We have now included the role of PARP1 in BER and biological significance. In addition, we have  provided detail explanation regarding the HMGB1 in the revised manuscript.

Reviewer 2 Report

This is a well-written paper by Zhao and colleagues summarizing the main milestones in the role of Base Excision Repair in innate immune cells and its relevance for cancer therapy.

Authors collected and reviewed recent literature evidence to demonstrate how BER inhibitors or aberrant DNA repair modulates innate inflammatory response and impact immunotherapy approaches. This review provided the substantial evidence to understand the impact of BER in innate immune response dynamics within the current immune based therapeutic strategy. Review is organized as follows: 1. Introduction; 2. Base excision repair in innate immune cells; 3. BER defect contributes in innate immune response; 4. Role of BER in macrophage plasticity; 5. Targeting PARP1 modulate macrophage mediated tumor inflammatory response; 6. Targeting APE1 modulate macrophage mediated tumor inflammatory response; 7. Exploiting BER as innate immune modulator for cancer therapy; 8. Exploiting DNA repair defect associated with reactivation of cGAS/STING in TME; 9. Inhibition of BER factors enhances program cell death ligand-1 (PD-L1) expression in cancer cells and 10. Conclusion and future direction. Accordingly, the authors provide the evidence that macrophages phenotypes and their functional plasticity are influenced by several factors including oxidative DNA damage and repair capacity. Subsequently accumulation of DNA damage leads to chromosomal instability and activate the innate immune signaling. Furthermore, dysregulation of BER factors in cancer cells can modulate the infiltration of innate immune cells to the tumor microenvironment. Interestingly, the authors indicated correlation of BER genes activity with M1/M2 macrophages polarization and suggested impact of BER activity on antitumor immunity. The authors extensively discussed latest literature data and its presentation is easy to follow for readers. Manuscript is illustrated with 4 figures explaining „Base excision repair capacity in macrophage.”; „Aberrant BER induces DNA sensor mediated inflammatory signaling”; „BER gene landscape and its impacts on macrophage infiltration in tumor microenvironment” and „BER based therapeutic intervention in immune cells (macrophage) modulate inflammatory signaling’. Illustration are very informative, helpful and provide great data presentation.

Paper has 192 references, including 5 self-citations (35, 67, 84, 88, 95) which are relevant to article's subject.

 This is an excellent review-study, and clinically valuable, especially for those researchers who were not familiar with the role of BER in innate immune response dynamics within the current immune based therapeutic strategy against cancer. This manuscript provide comprehensive information on this issue.

Minor comments:

Comment 1. Figure 3 and its relevant text paragraph appear in the manuscript before figure 2.  

Taken together, this paper by Zhao and colleagues represents a worthwhile contribution to the cancer research. I recommend the manuscript for further publication process.

Author Response

Comment 1. Figure 3 and its relevant text paragraph appear in the manuscript before figure 2.  

Taken together, this paper by Zhao and colleagues represents a worthwhile contribution to the cancer research. I recommend the manuscript for further publication process.

Response: We thank reviewer for encouraging comments and insightful positive response. We have corrected the figures legends and labels in the manuscript.

Reviewer 3 Report

Dear Authors,

The manuscript entitled "Role of Base Excision Repair in Innate Immune Cells and Its Relevance for Cancer Therapy" is of interest to the scientific community. It is well-written in general, but, it is necessary to put the final touches in some sentences. 

1. Additionally, there are some format issues, please see sections on page 3 for instance, and sections on pages 7 and 8-11, which have different line spacing. 

2. I suggest adding section numbering to enhance the readability of the manuscript

3. Since the manuscript has lots of abbreviations, I think that including a list of abbreviations will ease and support the reading of the non-expert readers. 

4. Figure 2: the description of the left and right panels are interchanged in the figure legend, please correct. 

5. Figure 3 has been deleted, but the legend is still in the manuscript. So, either include the figure or delete the legend (if it is deleted, please keep in mind that it is cited in the text and re-number the fourth figure). 

Author Response

Response to Reviewer 3

The manuscript entitled "Role of Base Excision Repair in Innate Immune Cells and Its Relevance for Cancer Therapy" is of interest to the scientific community. It is well-written in general, but, it is necessary to put the final touches in some sentences. 

1.Additionally, there are some format issues, please see sections on page 3 for instance, and sections on pages 7 and 8-11, which have different line spacing. 

Response:  We have corrected the line spacing issues and formatted in revised manuscript.

  1. I suggest adding section numbering to enhance the readability of the manuscript

Response: We have now added line numbers.

  1. Since the manuscript has lots of abbreviations, I think that including a list of abbreviations will ease and support the reading of the non-expert readers. 

Response: We have defined all the  abbreviations in the text of the manuscript.

  1. Figure 2: the description of the left and right panels are interchanged in the figure legend, please correct. 

Response: We have corrected the panels in Figure 2 legends.

  1. Figure 3 has been deleted, but the legend is still in the manuscript. So, either include the figure or delete the legend (if it is deleted, please keep in mind that it is cited in the text and re-number the fourth figure). 

Reviewer 4 Report

In this review Zhao and coll. described the role of BER in cancer and innate immune cells and its impact on innate immune signaling within tumor microenvironment. As a whole this work provides a clear and comprehensive view of the current state of the art  also providing articolated framework to develop therapeutic strategies. 

Author Response

Response to Reviewer 4

In this review Zhao and coll. described the role of BER in cancer and innate immune cells and its impact on innate immune signaling within tumor microenvironment. As a whole this work provides a clear and comprehensive view of the current state of the art  also providing articulated framework to develop therapeutic strategies. 

Response: We thank reviewer four for encouraging comments

Round 2

Reviewer 1 Report

The manuscript "Role of Base Excision Repair in Innate Immune Cells and Its Relevance for Cancer Therapy", written by Zhao S, Habib SL, Senejani A, Sebastian M and Kidane D, is a review describing the role of base excision repair in macrophages and the role of macrophages in cancer treatment.

Although the authors improved the first part of the manuscript, it still raises many questions.

There are still many grammatical and syntax mistakes.

Some statements are not correct or need references. Many statements are taken from the context and without explanation of the experimental settings, cell types or treatment cannot be used to approve the authors' hypotheses.

These are only some of the examples:

Already in the Abstract there is still a sentence "germline and somatic mutations are abundant in human cells" which needs an experimental prove.

In the paragraph "BER defect contributes to innate immune response" there are many statements that require explanations, such as "germline variant of BER genes".

Describing cGAS-STING system, ref 116 is mentioned and it does not describe that system. Ref. 118 talks about activation of the system as a reaction on the pathogen and that is not mentioned. Also, the conclusion in one of the reference articles indicates huge difference in response in normal and conditions of autoimmunity.

-There is no such thing as exogenous DNA damage.

-Second paragraph on page 8 is not clear.

-HMGB1 is released from necrotic cells

On page 9 there is description of PARP inhibitor "promoting DNA damage": PARP inhibition does not lead to repair, it does not cause the damage. And the referenced article describes BRCA- cells, and this is not mentioned.

Author Response

Response to Reviewer 1

Although the authors improved the first part of the manuscript, it still raises many questions.

There are still many grammatical and syntax mistakes.

Some statements are not correct or need references. Many statements are taken from the context and without explanation of the experimental settings, cell types or treatment cannot be used to approve the authors' hypotheses.

Response:  The revised manuscript is edited by professional English native language speaker.

These are only some of the examples:

Already in the Abstract there is still a sentence "germline and somatic mutations are abundant in human cells" which needs an experimental prove.

Response: We have modified the sentences and provide relevant references in the revised manuscript in the abstract section  as well as in the main section of the manuscript (line 28-32).

In the paragraph "BER defect contributes to innate immune response" there are many statements that require explanations, such as "germline variant of BER genes".

Response: We have modified  the sentence and included examples of germline variant BER genes that impact  genomic integrity and alter inflammatory response (line: 164-165 and 176-178).

Describing cGAS-STING system, ref 116 is mentioned and it does not describe that system. Ref. 118 talks about activation of the system as a reaction on the pathogen and that is not mentioned. Also, the conclusion in one of the reference articles indicates huge difference in response in normal and conditions of autoimmunity.

Response:  We have provided additional description of cGAS and STING have been identified as intracellular DNA sensors that activate the interferon pathway in response to virus infection and intercellular pathogens to defense against pathogens. We also provided additional references (97 and 116; line 228-230).

-There is no such thing as exogenous DNA damage.

Response:  We have now corrected “ exposure to exogenous DNA damaging agents(line 246)

-Second paragraph on page 8 is not clear.

Response: We have now modified paragraph two and present additional explanation  how genetic variants of BER genes may result in differences in immune system components including the activation states of immune-cell types and regulation of immune-related genes and have provided added additional references (line 299-312)

-HMGB1 is released from necrotic cells

Response: We have provided additional references including HMGB1 released from activated innate immune cells including monocytes and macrophage as well as necrotic cells (line:426-428).

On page 9 there is description of PARP inhibitor "promoting DNA damage": PARP inhibition does not lead to repair, it does not cause the damage. And the referenced article describes BRCA- cells, and this is not mentioned.

Response: We have now modified the sentencePARP1 inhibitor impairs DNA repair capacity and enhance accumulation of DNA damage and provide relevant references  (line 519-522).

Round 3

Reviewer 1 Report

The manuscript Zhao et al. does not have enough quality to be published in this form. To my opinion some facts are taken from the context, and without describing the context, cell type, conditions, they can mislead. I think many statements should therefore be checked.

Some basic abbreviations are not explained properly (such as TLR: Toll like receptor, not Tall like receptor – in all 3 variants of the manuscript) or DAMP (damage.. not dangerous).

There are still many grammatical mistakes.

Here are only some examples, as illustration:

"For instance, activated macrophage induced ROS and RNI caused DNA damage including 8-nitroguanine and 8-oxo-7,8-dihydro-2'-deoxyguanosine (8-oxodG)."

-it is not clear in which cells

"Furthermore, cells harbor SSBs progress into S-phase encounter DNA-replication apparatus and convert to double-strand breaks (DSBs)"

-which cells, in which conditions?

"Those DNA damaged site will likely release DNA fragments into cytosolic compartment of the cells and potentially would be recognized as DAMP to trigger innate immune response"

in the reference cited, experiments were done on lupus erythematosus cells, so, these processes cause autoimmunity – therefore are not expected in normal cells, but those lacking one enzyme.

"Macrophage has the ability to activate intrinsic cGAS/STING pathways due to persistence of endogenous oxidative stress associated  DNA base damage and promote pro-inflammatory response (129)."

The citing reference (ref. 129) does not write about cGAS/STING.